# Realization of all-band-flat photonic lattices

Jing Yang[1,2,3], Yuanzhen Li ®[2,3], Yumeng Yang[2,3], Xinrong Xie ®[2,3], Zijian Zhang[2,3], Jiale Yuan[1], Han Cai ®[1,4], Da-Wei Wang ®[1,4,5] ✉ & Fei Gao ®[1,2,3] ✉

Flatbands play an important role in correlated quantum matter and have promising applications in photonic lattices. Synthetic magnetic fields and destructive interference in lattices are traditionally used to obtain flatbands. However, such methods can only obtain a few flatbands with most bands remaining dispersive. Here we realize all-band-flat photonic lattices of an arbitrary size by precisely controlling the coupling strengths between lattice sites to mimic those in Fock-state lattices. This allows us to go beyond the perturbative regime of strain engineering and group all eigenmodes in flatbands, which simultaneously achieves high band flatness and large usable bandwidth. We map out the distribution of each flatband in the lattices and selectively excite the eigenmodes with different chiralities. Our method paves a way in controlling band structure and topology of photonic lattices.

Flatbands lead to localization, high density of states (DOS), and non-trivial topology, attracting increasing interest in electronic materials[1–4], atomic physics[5–10], and photonic lattices[11–23]. In strongly correlated electronic systems[24,25], the small energy width and high DOS of flatbands facilitate the observation of many-body physics such as fractional quantum Hall effect[26], ferromagnetism[27–29], and superconductivity[30–34]. Flatbands also have promising applications in photonic systems[35–37]. The zero group velocity in flatbands can be used to achieve slow light[38,39], enhanced light-matter interaction[40], and dispersionless image transmission[41,42]. Systematic methods have been developed in generating flatbands in one[43,44] and two[45] dimensions. In particular, by carefully engineering the hopping strengths between lattice sites, it is possible to realize all-band-flat (ABF) lattices[46–49], which can balance the trade-off between flatness of the bands and the useful bandwidth[35], by turning all bands flat to utilize the full lattice energy spectra. Such ABF lattices also provide a unique platform to investigate Aharonov-Bohm caging[1], compact localized states[50], non-linear and quantum caging[46,47]. Interestingly, it has been theoretically proposed that by finely tuning the coupling strengths we can obtain finite ABF lattices[51,52]. However, limited by the achievable range of the coupling strengths and the dissipation of the resonators, so far not all eigenstates can be grouped into flatbands in experiments[53,54].

In this Letter, we experimentally realize ABF honeycomb lattices of microwave resonators by engineering the coupling strengths between resonators to mimic the Fock-state lattices (FSLs) of a three-mode Jaynes-Cummings (JC) model, an emulation of quantum bosonic topological states[55,56] in photonic lattices. We precisely control the coupling strengths at different locations and group all eigenstates in flatbands, such that high DOS is obtained at discrete energies. The perturbative strain field due to the spatially varying coupling strengths introduces a pseudo-magnetic field, which has been used to generate a few flatbands near the Dirac points[53,54,57–61]. Here we go beyond the perturbative regime of the strain engineering to realize ABF lattices. We measure the distribution of states and manage to selectively excite different eigenstates in a flatband. Our results validate a scalable method to generate ABF lattices with arbitrary size and offer a platform to study topological transports in photonic lattices.

## Results
### Simulating FSL flatbands with photonic lattices
Electromagnetic resonators and waveguides have been widely used to simulate topological physics of electrons[62]. Topological edge modes propagating unidirectionally without being scattered by local defects are promising for applications in robust photonic devices[48,63,64].

[1]Zhejiang Province Key Laboratory of Quantum Technology and Device, School of Physics, and State Key Laboratory for Extreme Photonics and Instrumentation, Zhejiang University, Hangzhou, China. [2]ZJU-Hangzhou Global Science and Technology Innovation Center, College of Information Science and Electronic Engineering, Zhejiang University, Hangzhou, China. [3]International Joint Innovation Center, Key Laboratory of Advanced Micro/Nano Electronic Devices & The Electromagnetics Academy at Zhejiang University, Zhejiang University, Haining, China. [4]College of Optical Science and Engineering, Zhejiang University, Hangzhou, China. [5]CAS Center for Excellence in Topological Quantum Computation, University of Chinese Academy of Sciences, Beijing, China. ✉e-mail: dwwang@zju.edu.cn; gaofeizju@zju.edu.cn

Beyond classical topological photonics, the Fock states of light form strained lattices with ABF energy spectra[55], which have been experimentally realized in a superconducting circuit[56]. Such quantum topological states of bosonic modes provide new tools to design classical photonic lattices for flat-band optical engineering. We note that one-dimensional photonic lattices that mimic the coupling between Fock states for coherent and topological transport have been proposed[65,66] and experimentally realized[67,68]. Here we extend the technique to two dimensions to realize ABF lattices.

We use a honeycomb lattice of microwave resonators with site-dependent coupling strengths (see Fig. 1a)[55] to obtain the ABF energy spectrum. The resonators are labelled by $A_{ijk}$ and $B_{ijk}$ for A and B sublattices, with $i, j,$ and $k$ being the indices in $\mathbf{e}_1 = (-\sqrt{3}/2, -1/2)$, $\mathbf{e}_2 = (\sqrt{3}/2, -1/2)$ and $\mathbf{e}_3 = (0, 1)$ directions, satisfying $i + j + k + (\xi + 1)/2 = N$ with $\xi = -1$ and 1 for A and B sites, respectively. At the triangular lattice boundary one of $i, j, k$ becomes zero, corresponding to the vacuum state in FSLs. In total, the honeycomb lattice contains $(N+1)^2$ sites. The coupling strength between $A_{ijk}$ and $B_{i-1jk}$ site is $\sqrt{i}t_0$, and the same rule applies to $j$ and $k$. The square root factors are introduced to emulate the couplings between different harmonic states[55,56,69–71], which involves the properties of bosonic annihilation operators. In the lattice the coupling strengths vary from $t_0$ to $\sqrt{N}t_0$, in a range smaller than the existing proposals[51,52] by a factor of $\sqrt{N}$, which facilitates the following experimental realization. These couplings are described by the tight-binding Hamiltonian,

$$H = t_0 \left[ \sum_{i,j,k} \left( \sqrt{i}b_{i-1jk}^\dagger + \sqrt{j}b_{ij-1k}^\dagger + \sqrt{k}b_{ijk-1}^\dagger \right) a_{ijk} + h.c. \right], \quad (1)$$

where $a_{ijk}$ and $b_{ijk}$ are the annihilation operators of $A_{ijk}$ and $B_{ijk}$ resonators. The eigenenergies are solved analytically (see Supplementary Section I.A),

$$E_m = \pm \sqrt{3m}t_0, \quad (2)$$

with $m = 0, 1, 2, \ldots, N$ and degeneracies $N - m + 1$. Therefore, we group all eigenstates in $N + 1$ flatbands.

In order to achieve wide tuning range of the coupling strengths while maintaining narrow linewidths, we construct such lattices using aluminum coaxial cavities shown in Fig. 1b, whose hexapole mode of transverse magnetic (TM) polarization has a resonant frequency 12.002 GHz. This TM mode is confined inside the cavity as shown in Fig. 1b, c, without evanescent fields in the ambient[55]. We employ short waveguides (WGs) to couple each resonator with its three neighbors. The three coupling WGs with widths $d_1, d_2,$ and $d_3$ are connected to three openings on the resonator wall. To correct the slight frequency shift due to these openings, we use a fourth opening with width $d_4$ to align resonance frequencies of all cavities. All these widths are individually designed for each resonator to obtain the ad-hoc engineered coupling strengths. The evanescent waves in the WGs couple neighboring resonators with coupling strengths being determined by the widths of the WGs. Numerical simulation of the frequency splitting of two connected resonators shows the relation $t = (4.4d^2 - 37.8d + 89.8)$ MHz (see Fig. 1d, Supplementary Section II). We individually design the widths to achieve the required coupling strength distribution beyond perturbative regime. The minimum coupling strength $t_0 = 69$ MHz is much larger than the resonance

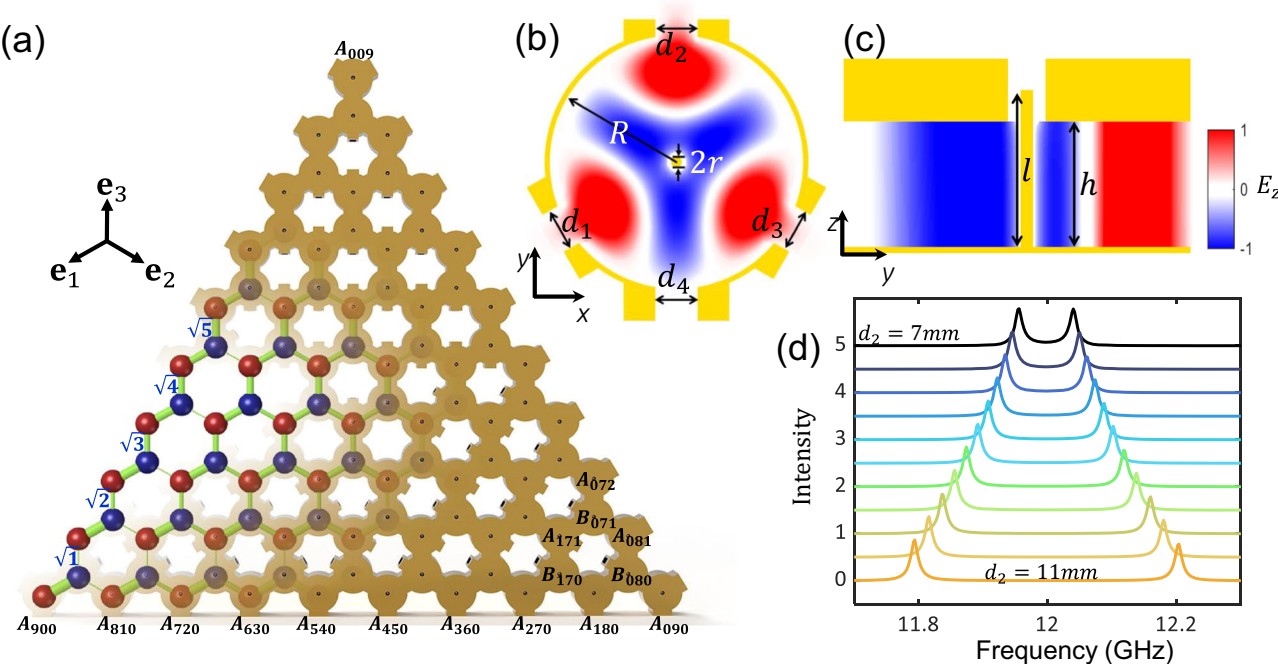

**Fig. 1 | Photonic lattice mimicking the coupling of Fock-state lattices. a** An ABF honeycomb lattice of microwave resonators with $N = 9$, containing 100 resonators. Red and blue sites denote A and B resonators connected by lines with widths proportional to the local coupling strengths. **b, c** The geometry of a single cavity and the $E_z$ field distribution of the TM mode in the $xy$ **b** and $yz$ **c** planes. The cavity has an inner radius $R = 24$ mm and a height $h = 20$ mm. An aluminum rod in the center of the cavity (with radius $r = 1$ and height $l = 25$ mm) is used to make contact with the antenna to maintain stability in excitation and measurement. Each resonator is coupled to three adjacent resonators via short waveguides with widths $d_1, d_2,$ and $d_3$. The distance between two adjacent resonators is 50 mm. An extra opening with width $d_4$ is used to tune the resonance frequency. The radius of the hole in the top of the cavity is 3 mm. **d** Numerical simulation (see Supplementary Section II) of the frequency splitting of two coupled resonators. The coupling channel width $d_2$ changes from 7 mm to 11 mm with a 0.4 mm step, while keeping $d_1 = d_3 = 8$ mm.

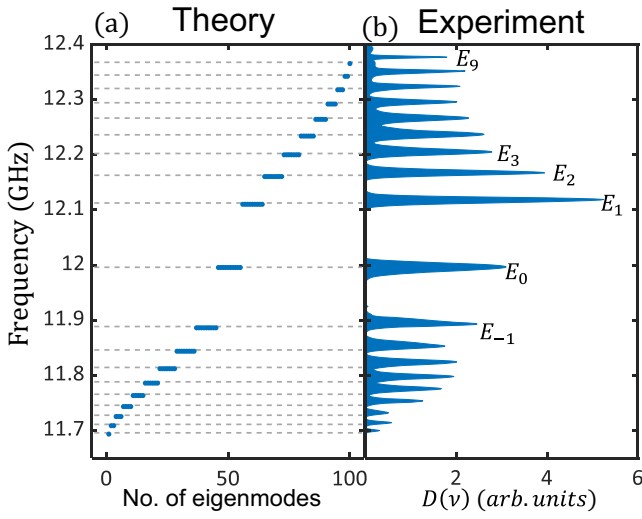

**Fig. 2 | Density of states of the flatbands. a** The numerically simulated eigen-energies for the lattice with $N = 9$, $t_0 = 69$ MHz, and next-nearest-neighbor coupling $\kappa = 2$ MHz. **b** Experimentally measured $D(\nu)$, which is obtained from reflection spectra of each resonator by vector network analyzer 3672C. The peaks correspond to the 19 flatbands with discrete eigenenergies. The positions of the peaks coincide with the theoretical prediction. The spectra weight indicates the degeneracy of each band.

linewidth ($\gamma = 10$ MHz) of a single resonator, such that the discrete flatband energy levels are completely separated from each other.

## Discrete energies of flatbands

We experimentally characterize the flatbands by measuring reflection spectra site by site. We employ an antenna which functions as a point source and a detector simultaneously. The antenna connected to a vector network analyzer is inserted through the top hole on cavities, and contacts the inside central rod. We measure the reflection spectrum on each site, $R(\mathbf{r}_j, \nu)$, which are related to the local DOS $D(\mathbf{r}_j, \nu)$ (see ref. 72, Supplementary Section III),

$$
\begin{aligned}
D(\mathbf{r}_j, \nu) &\equiv \sum_m \frac{2\gamma}{(\nu - \nu_m)^2 + \gamma^2} |\Phi_m(\mathbf{r}_j)|^2 \\
&\propto 1 - |R(\mathbf{r}_j, \nu)|^2,
\end{aligned} \tag{3}
$$

where $\Phi_m(\mathbf{r}_j)$ represents the $m$th lattice mode of eigenfrequency $\nu_m$ and $\mathbf{r}_j$ denotes the position of the $j$th resonator. The total DOS can be evaluated with $D(\nu) = \sum_j D(\mathbf{r}_j, \nu)$.

We measure the reflection spectra at all resonators, and obtain $D(\nu)$ shown in Fig. 2b. The result features with peaks corresponding to the 19 flatbands of the lattice shown in Fig. 2a. The measured zeroth Landau level $E_0$ located at $\omega_0 = 11.997$ GHz, slightly deviating from the frequency of a single resonator. Such deviation is due to next-nearest-neighbor couplings (see Supplementary Section I.B). The peaks above and below $E_0$ correspond to the positive and negative Landau levels, respectively. The frequency difference between $E_0$ and $E_1$ is 0.12 GHz, consistent with the theoretical value $\sqrt{3}t_0$. The spectra weight (peak area) reflects the degeneracy of the corresponding Landau level, consistent with the theoretical prediction. The positive Landau levels show larger band splitting than the negative ones due to next-nearest-neighbor couplings induced by a higher-order cavity mode at 12.5 GHz (see Supplementary Section I.B). The positive Landau levels exhibit larger spectra weights than the negative ones due to indirect couplings between adjacent resonators (see Supplementary Section I.C).

## Spatial distribution of modes in flatbands

We further experimentally image the modes in flatbands, by measuring reflection coefficients $R(\mathbf{r}_j, \nu_m)$. Figure 3 presents the simulated and captured mode patterns of $m$th flatband with $m = 0, 1, 6, 8, 9$ respectively. The pattern of the zeroth $m = 0$ Landau level in Fig. 3 shows anomalous parity, manifesting as nonzero local DOS only in the A sublattice. Such sublattice polarization is due to the chiral symmetry breaking, originating from the site number difference between the two sublattices. Different from the unstrained lattices where the zero-energy modes occupy the terminating A sites[53], here the zero-energy modes are confined within the incircle of the lattice. This is because the nonperturbative strain induces a semimetal-insulator phase transition on the incircle. Within the incircle, the energy bands touch at strain-shifted Dirac points, while outside of the incircle, a band gap is opened. Therefore, the zero-energy modes only exist within the incircle, which is a Lifshitz topological edge[55].

By evaluating the variances of the mode functions (see Supplementary Figs. S4 and S5), we observe that the mode functions spread from the center to the edges when $m$ increases from 0 to $N/2$, and then shrink to the center when $m$ increases from $N/2$ to $N$. Their spatial distributions remain $C_3$ symmetry with respect to the center of the lattice. Different Landau levels have their own preferred locations to occupy. Regarding $m = 1$, the eigenmodes on the A sites have high weights in the three corners of the lattice, while for $m = 6$, the three edges are preferred. The 8th Landau level has an annulus distribution with zero intensity in the center, and the 9th Landau level which contains only one mode is localized near the center. The eigenmodes in higher Landau levels occupy sites beyond the incircle, with nearly equal populations in the two sublattices (see Fig. 3). The distinctive distributions of the Landau levels give us an additional controlling knob to selectively excite a Landau level at specific positions of the lattice. Such a feature can help us to use the whole energy spectra for flatband engineering. The precision in tuning the coupling strengths allows us to obtain a higher than 0.85 fidelity for most Landau levels (see the definition of fidelity and evaluation of the band flatness in Supplementary Section IV).

## Selective excitation of degenerate eigenmodes

In each Landau level, degenerate eigenmodes differentiate themselves with different chiralities $C$[55] (see details in Supplementary Section I), which plays the role of lattice momentum in an infinite lattice. The chiralities of degenerate states manifest as relative phase differences between lattice sites. For instance, the 7th Landau level has only 3 eigenmodes of chirality $C = 0$ and $\pm 2$ (see the field distributions in the middle of Fig. 4). The mode of $C = 0$ exhibits a $\pi$ phase change along the radial direction, but remains in phase along the angular direction. The eigenmode of $C = 2$ distributes far away from the center of the lattice, and has $4\pi$ phase change along the counterclockwise direction, similar to that of a vortex.

To selectively excite an eigenmode of specific chirality in a Landau level, we utilize multiple antennas with relative phase difference $\phi$ which matches the phase distribution at corresponding lattice sites. For $\phi = 0$ we excite the three sites with the same phase. For $\phi = 2\pi/3$ we excite the three sites with phases 0, $4\pi/3$ and $2\pi/3$ in the counterclockwise direction. The eigenmode with $C = 0$ can be efficiently excited with $\phi = 0$ near the center, but not for $\phi = 2\pi/3$. This is because the three sites have the same phase in this eigenmode. In contrast, for $C = 2$ the eigenmode can be only efficiently excited for $\phi = 2\pi/3$ away from the center, but not for $\phi = 0$, consistent with the corresponding phase distribution. We note that when the lattice sites are efficiently excited, the phase distributions in the lattice are in accord with those of the corresponding eigenmodes.

## Discussion

In conclusion, we construct two-dimensional ABF photonic lattices by mimicking the coupling strengths in FSLs. Strained hexagonal lattices

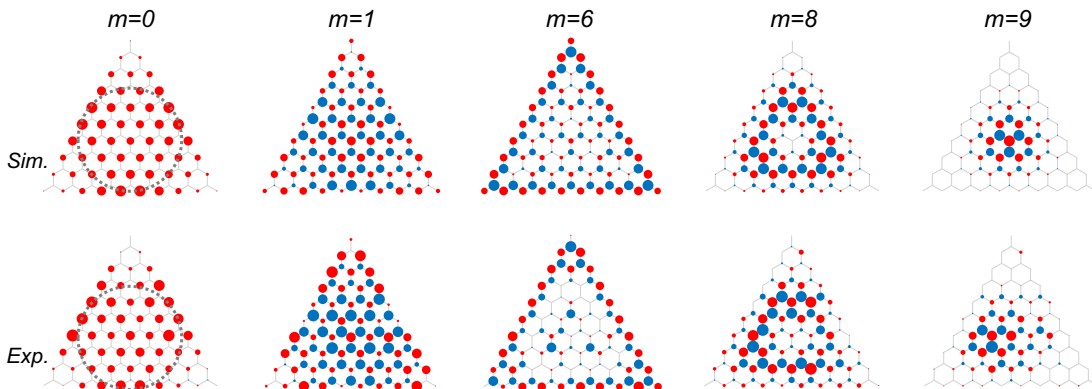

**Fig. 3 | Eigenmode distributions in the $m$th Landau levels with $m = 0, 1, 6, 8, 9$.** The first row shows the numerical simulation and the second row shows the experimental data. The radii of the circles are proportional to $D(\mathbf{r}_j, v)$ on the A sites (red) and B sites (blue). The zeroth Landau level only occupies A sites, while other Landau levels have approximately equal weights in the two sublattices. The eigenmodes in the zeroth Landau level are confined within the incircle (dashed lines), which separates the inside semi-metallic phase from the outside insulator phase.

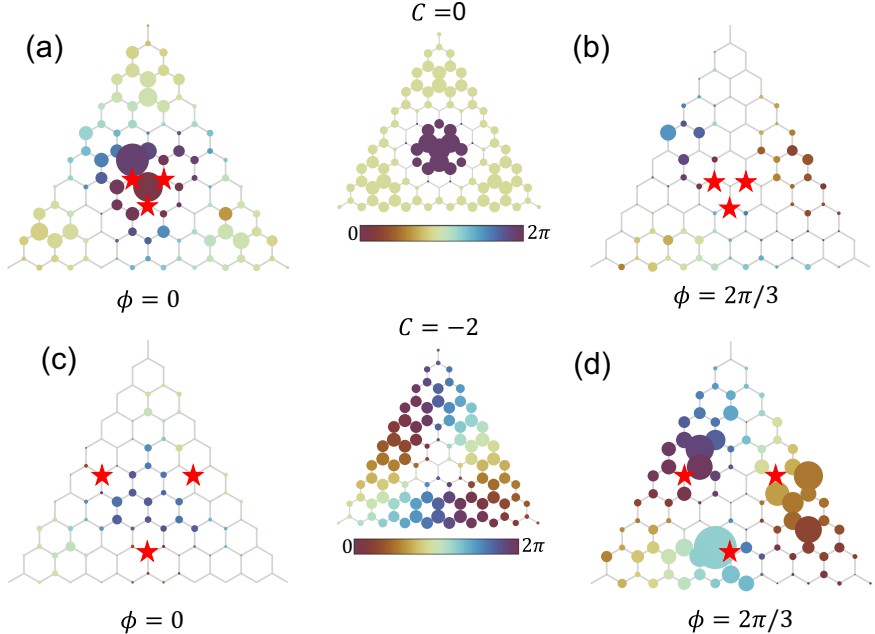

**Fig. 4 | Selective excitation of the eigenmodes in the 7th flatband. a, b** Field distribution with three excitation sites near the center, which have a large overlap with the eigenmode $C = 0$. **c, d** Field distribution with three excitation sites that have a large overlap with the eigenmode $C = 2$. The excitation phase difference $\phi = 0$ in **a**, **c**, and $\phi = 2\pi/3$ in **b**, **d**. The radii of the colored dots are proportional to the field intensity on the lattice sites, with colors indicating the phases. The two figures in the middle are the numerically simulated field distributions of the corresponding eigenmodes.

have been extensively used to synthesize pseudo magnetic fields, resulting in flat Landau levels[58–61]. However, only the first several Landau levels in the linear dispersion region can be obtained[53,54]. To fully exploit flatbands for photonic engineering, this approach goes beyond the perturbative strain engineering regime to obtain ABF spectra. The method can be generalized to the rich configurations of FSLs of atom-cavity coupled systems[73], which are characterized by ABF spectra with large degeneracy. Compared with the existing proposals for ABF lattices with discrete translational symmetry[46,47], our approach is scalable for arbitrary-size lattices without non-flat edge modes. By adding nonlinear elements e.g. varactor diodes in the resonators[74,75], we can introduce Kerr nonlinearity and investigate the nonlinear localization effect and other many-body effect in such lattices[46,76–78].

With realizable Kerr nonlinearity, breathing dynamics[77] between flat-bands can be observed in the current setup (see simulation in Supplementary Section VII). Besides, it has been proved that flatbands can enhance the second harmonic generation[79]. We can introduce second order nonlinearity in our current setup[80,81]. With all bands being flat, we expect such an effect can be further enhanced. The current approach can be applied to ABF photonic waveguides, which have promising applications in dispersionless imaging[41], nonlinear polaritons[82] and topological solitons[83]. In particular, the one-dimensional version of similar photonic lattice engineering has been realized in the topological transport of light field[68]. Such finite-size ABF lattices can also be used in nano-lasers[84–86] with high frequency purity and flexible mode configurations.

## Methods

### Simulation

All the simulations are conducted with CST Microwave Studio. In these simulations, the metal under microwave frequency is modeled as a perfect electrical conductor (PEC). The discrete ports are utilized as the excitation sources. The simulation regions are slightly larger than the objects under study, and are enclosed with boundaries of open space.

### Experimental setup

All the metallic cavity resonators are made of 6061 aluminum, and are fabricated with Computer Numerical Control (CNC) technologies. All the measurements are carried out on Ceyear-3672C Vector network analyzer. Monopole antennas of 3-mm length are employed for excitation and detection. In selective excitation experiments, the lattices are excited simultaneously by three monopole antennas which are of the same amplitude but different phases. Adjustable attenuators (KST-30) and phase shifters are used to ensure the three antennas have the same amplitude and desired phase differences respectively.

## Data availability

The data generated in this study have been deposited in Figshare database under the following accession code https://doi.org/10.6084/m9.figshare.25027172.v2.

## Code availability

The code that supports the plots within this paper can be found in Figshare database under the following accession code https://doi.org/10.6084/m9.figshare.25027172.v2.

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

## Acknowledgements

This work was supported by the National Key Research and Development Program of China (Grants No. 2022YFA1404902 (F.G.), 2019YFA0308100 (D.W.W.)), the National Natural Science Foundation of

China (Grants No. 62171406 (F.G.), 12325412 (D.W.W.), 11934011 (D.W.W.), U21A20437 (H.C.)), ZJNSF under Grant No. Z20F010018 (F.G.), the Strategic Priority Research Program of the Chinese Academy of Sciences under Grant No. XDB28000000 (D.W.W.), and the Fundamental Research Funds for the Central Universities under Grant No. 2023QZJH13 (D.W.W.).

## Author contributions

D.W.W. and F.G. conceived the project. J.Y., Y.L., and F.G. designed and constructed the sample. J.Y., J.L.Y., and H.C. performed numerical simulations. J.Y. carried out the experiment and collected data. Y.Y., X.X., and Z.Z. assisted the measurement. All authors analyzed and discussed the results. J.Y., D.W.W., and F.G. wrote the manuscript with inputs from all other authors.

## Competing interests

The authors declare no competing interests.
