## [Peer Review File · Nature Communications]

REVIEWER COMMENTS

Reviewer #1 (Remarks to the Author):

The authors propose an experimental realisation of an all-bands-flat honeycomb lattice realised as an array of microwave resonators with site and direction dependent hoppings.

They compare experimental spectrum and the numerical results of the flat levels (Landau levels), which show good agreement.

The authors also discuss how eigenstates of individual levels can be excited, providing experimental results supplemented with numerical simulations.

This provides robust evidence for the realisation of the all-bands-flat network.

Overall the manuscript is convincing, well written (the few remarks are listed below) and reads well.

Networks with all bands flat are promising platforms for implementing interesting phenomena, like nonlinear caging, as is already mentioned in the text.

On the other hand their experimental realisations are tricky due to strict requirements for the hoppings.

Therefore I think robust implementation of such a network is an important achievement worth of publication in Nature Communications.

Questions

There are two issues that I would like the authors to clarify:

1. The authors mentioned in the introduction, that:

"The strain field due to the spatially varying coupling strengths introduces a pseudo-magnetic field, which has been used to generate a few flatbands near the Dirac points [51, 52, 55–60].

Here we go beyond the perturbative regime of the strain engineering to realize ABF lattices"

Are the authors aware of the "PRL 117, 266801 (2016) - Strain-Induced Landau Levels in Arbitrary Dimensions with an Exact Spectrum",

where perfectly flat Landau levels were designed using linearly increased strain on finite clusters?

I believe the model (for the honeycomb lattice) from that letter coincides with the one considered by the authors.

How the result of the Letter compare to the results presented in the manuscript?

2. Could the authors clarify the elusiveness in experimental realisations of the model, e.g.:

"However, the experimental realization of all-flat bands in a finite two-dimensional lattice is still elusive [51, 52]."

The authors of Ref. 51 also consider coupled microwave resonators as an implementation of the strained honeycomb lattice, and also seem to observe flat bands.

Remarks

1. Eq. 3: the quantity $S_{\{11\}}$ has not been defined in the text
2. p.3, right column, 2nd paragraph from the top, last sentence: I presume "fidelity of the Landau level" quantifies its flatness?

I would suggest to define it explicitly

3. Fig. 4: "insets" -- I presume the insets are the plots in the middle? It might be better to refer to them as simply "the plots in the middle"?

Reviewer #2 (Remarks to the Author):

In the manuscript "Realization of all-band-flat photonic lattices", Yang et al. investigate the experimental realization with microwave resonators of a lattice model with only flat bands, which corresponds to a strained honeycomb lattice beyond the perturbative regime in a Fock-space lattice (FSL). While in FSLs the ad-hoc space-dependent couplings naturally arise via the bosonic statistics of the creation/annihilation operators, here the authors employ the tunability of the platform to exactly match those couplings for the resonators. They observe flat bands and also some properties of the eigenstates (their chirality or phase winding).

The manuscript is scientifically sound and the results are convincing. However, I believe that this work is suited to a more specialized journal for the following reasons:

- 1) on the theoretical side, the authors are realizing a linear system whose physics is relatively simple and known. Besides, many other models with flat bands have been realized over the last years, so I do not find a particular advance here, except that this work provides the realization of yet another model with all bands flat, an extremely interesting example of suppressed transport distinct from AB caging. Thus, I find the results incremental in the respect of many realizations of flat band systems.

2) on the experimental side, the realization of the model does not seem to require particular challenging steps, techniques or innovation, but standard engineering methods for this platform. I therefore fail to find a nontrivial experimental advance. Besides, while the needed couplings arise naturally with harmonic oscillator systems here they require ad-hoc engineering, thus wondering if there is a need to use this platform if no other advantage (besides the possibility to realize large lattices) is possible. If the authors were able to provide arguments that these results can hardly be seen in most of the other platforms (like photonic waveguides or ring resonators, where large systems and programmable couplings can also be designed with relatively small experimental effort), thus explaining the need or unique advantages of this platform, or if they demonstrated interesting nonlinear effects on such flat bands, this would be a quite different story.

Here a couple of comments:

- I could not find a fully clear explanation of how the coupling strength is experimentally tuned as \sqrt{i} . Is it done by changing the widths d_i or by other means? I suppose that the sentence "Three openings with individually designed widths.." below eq. 2 is the answer to my doubt. If that's the case, I suggest to be explicit and indicate that the individual design of the openings is designed to provide the \sqrt{i} couplings in order to avoid that a reader who is not completely familiar with the platform has to deduce this on their own.

- At the end of the manuscript, the authors indicate the possibility of exploring nonlinear effects by adding a nonlinear medium. As the authors are not citing any reference for this, is it experimentally possible and can this lead to sizeable nonlinear effects? Can they cite experimental literature showing such potential of nonlinear physics with this platform.

- The authors mention nonlinear caging as an outlook. However, caging (even in the nonlinear regime) typically involves models that have all-bands-flat dispersions due to interference in the linear regime and special connectivities of the lattice, namely different sublattices having different connectivities. I therefore do not see why the model studied in this work would lead to non-linear caging effects, in the same way as Landau levels are not known, to the best of my knowledge, to have the property of non-linear caging. If the authors are making a conjecture, or have in mind literature in this direction, they should explain what reasonings have they made to believe that such phenomenology of non-linear caging would occur here. Otherwise they should remove a claim of nonlinear caging. Besides, nonlinear caging was initially studied in the following articles that the authors are missing from their references:

Nonlinear symmetry breaking of Aharonov-Bohm cages

Goran Gligorić, Petra P. Beličev, Daniel Leykam, and Aleksandra Maluckov

Phys. Rev. A 99, 013826 – Published 15 January 2019

Nonlinear dynamics of Aharonov-Bohm cages

Marco Di Liberto, Seabrata Mukherjee, and Nathan Goldman

Phys. Rev. A 100, 043829 – Published 22 October 2019

Is there any other interesting nonlinear effect that could be explored in this model if caging is ruled out? I think the outlook part is a bit vague and may benefit from a clearer perspective on the impact of this work.

- I take the opportunity to stress that tight-binding models with couplings dependent on the \sqrt{n} , where n is the Fock-state occupation, may arise in general setups that exploit harmonic oscillator states as a synthetic dimension, besides the JC models that the authors are referring to. For example, here some relevant works in this direction that the authors could cite

Synthetic dimensions for cold atoms from shaking a harmonic trap

Hannah M. Price, Tomoki Ozawa, and Nathan Goldman

Phys. Rev. A 95, 023607 – Published 9 February 2017

Quantized Hall Conductance of a Single Atomic Wire: A Proposal Based on Synthetic Dimensions

G. Salerno, H. M. Price, M. Lebrat, S. Häusler, T. Esslinger, L. Corman, J.-P. Brantut, and N. Goldman

Phys. Rev. X 9, 041001 – Published 1 October 2019

Photonic topological insulator in synthetic dimensions

Eran Lustig, Steffen Weimann, Yonatan Plotnik, Yaakov Lumer, Miguel A. Bandres, Alexander Szameit & Mordechai Segev

Nature volume 567, pages356–360 (2019)

Resubmission of NCOMMS-23-15161A "Realization of all-band-flat photonic lattices"

by J. Yang, Y. Li, Y. Yang, X. Xie, Z. Zhang, J. Yuan, H. Cai, D. W. Wang, and F. Gao
(Dated: November 13, 2023)

Summary of changes:

Note: These changes are also highlighted in red color in the revised manuscript and SM.

- We added Jiale Yuan as a coauthor because of his contribution in the extensive revision of the manuscript.
- In response to comment **Reviewer A.2 and Reviewer A.3**, we rewrote the introduction to emphasize our advances compared with previous theoretical and experimental works.
- In response to comment **Reviewer A.4**, we changed the notion $S_{11}(\vec{r}_j, \nu)$ to $R_{11}(\vec{r}_j, \nu)$ and defined it in SM.
- In response to comment **Reviewer A.5**, we defined fidelity in SM.
- In response to comment **Reviewer A.6**, we changed the "insets" to "the plots in the middle" in the main text.
- In response to comment **Reviewer B.2**, we emphasized the theoretical difference between ours and other works in the main text, and compared our scheme and previous works to achieve flatbands in SM.
- In response to comment **Reviewer B.3**, we emphasized the advantage of our system and experiment realization.
- In response to comment **Reviewer B.4**, we made simulation on realizing 3D lattices, coupling strength modulation and nonlinear effect in SM.
- In response to comment **Reviewer B.5**, we added the description on controlling the coupling strengths in our experiment.
- In response to comment **Reviewer B.6**, we cited the previous works on introducing nonlinear effects in similar systems. The feasibility of nonlinear effects in our setup is discussed in the SM. We added a list of previous works on the nonlinear effect in SM.
- In response to comment **Reviewer B.7**, we cited the mentioned works of nonlinear caging in the main text. We also numerically simulated the nonlinear breathing effect of our system in SM.
- In response to comment **Reviewer B.8**, we discussed the second harmonic generation by flatbands in the outlook.
- In response to comment **Reviewer B.9**, we properly cited the mentioned works in the main text.
- We changed the colorbar of Fig.4 in the main text.

Reply to reviewer A

Reviewer A.1: The authors propose an experimental realisation of an all-bands-flat honeycomb lattice realised as an array of microwave resonators with site and direction dependent hoppings. They compare experimental spectrum and the numerical results of the flat levels (Landau levels), which show good agreement. The authors also discuss how eigenstates of individual levels can be excited, providing experimental results supplemented with numerical simulations. This provides robust evidence for the realisation of the all-bands-flat network. Overall the manuscript is convincing, well written (the few remarks are listed below) and reads well. Networks with all bands flat are promising platforms for implementing interesting phenomena, like nonlinear caging, as is already mentioned in the text. On the other hand, their experimental realisations are tricky due to strict requirements for the hoppings. Therefore I think robust implementation of such a network is an important achievement worth of publication in Nature Communications.

Reply: We thank the Referee for carefully reading our manuscript and judging that our results are convincing, well written and worth of publication in Nature Communications.

Reviewer A.2: #Questions: There are two issues that I would like the authors to clarify: 1. The authors mentioned in the introduction, that: "The strain field due to the spatially varying coupling strengths introduces a pseudomagnetic field, which has been used to generate a few flatbands near the Dirac points [51, 52, 55–60]. Here we go beyond the perturbative regime of the strain engineering to realize ABF lattices" Are the authors aware of the "PRL 117, 266801 (2016) - Strain-Induced Landau Levels in Arbitrary Dimensions with an Exact Spectrum", where perfectly flat Landau levels were designed using linearly increased strain on finite clusters? I believe the model (for the honeycomb lattice) from that letter coincides with the one considered by the authors. How does the result of the Letter compare to the results presented in the manuscript?

Reply: We thank the Referee for raising this question and providing this reference. In the mentioned paper, the authors theoretically proposed a flatband model in arbitrary dimensions using linearly increased strain on finite lattices. As far as we know, no experimental implementation of this model has been reported. For a given lattice size N , their model requires that the coupling strengths can be tuned from t_0 to Nt_0 , which is larger than our changing range, from t_0 to $\sqrt{N}t_0$, and thus more difficult to realize. Of course, it is still possible to realize the mentioned proposal in our experimental setup, but for a definite changing range of the coupling strength, the realizable size of the lattice is smaller than our proposal, by a factor of N in the number of lattice sites (by a factor of \sqrt{N} in the length scale of the lattice).

Reviewer A.3: Could the authors clarify the elusiveness in experimental realizations of the model, e.g.: "However, the experimental realization of all-flat bands in a finite two-dimensional lattice is still elusive [51, 52]." The authors of Ref. 51 also consider coupled microwave resonators as an implementation of the strained honeycomb lattice, and also seem to observe flat bands.

Reply: As pointed out by the Referee, The original Ref. [51] (Ref. [53] in the revised version) also engineered the hopping strength to achieve a couple of flatbands. But the challenge lies in how to make all energy bands flat. The scheme in the original Ref. [51] still relies on strain engineering by changing the distances between resonators, which can be regarded as an effective magnetic field near the Dirac points. The tight-binding approximation requires that such dielectric resonators are of high quality factors (Opt. Lett. 24, 711 (1999)), and are coupled through evanescent waves. In such a scheme, a large inter-resonator distance results in negligible evanescent coupling, while a small distance leads to the breakdown of tight-binding conditions. Therefore, it is challenging to engineer the coupling strength by tuning the distance in very limited space, especially beyond the perturbative regime. However, to realize all-band-flat energy spectra, we need more drastic change of the strain, while the tight confinement of evanescent waves sets limits in the tuning range and precision of the coupling strengths. In the original Ref. [51] and related works (M. Bellec, U. Kuhl, G. Montambaux, and F. Mortessagne, Phys. Rev. B 88, 115437 (2013)), the linewidth of the mode is 5 MHz while the coupling strength can be changed from 20 to 80 MHz by adjusting the distance between the resonators (10 mm - 15 mm). The tunable range is restricted by the allowed geometry and sophisticated design on the configuration of the whole lattice is required. In our experiment, we tune the coupling strengths by changing the widths of the connecting waveguides between two resonators without deforming the lattice. With mode linewidth 10 MHz, the coupling strength can be tuned from 20 to 400 MHz without changing the lattice configuration. These advances allow us to go beyond the perturbative regime of strain engineering and obtain all-band-flat energy spectra.

Reviewer A.4: #Remarks:1. Eq. 3: the quantity S_{11} has not been defined in the text.

Reply: We thank the Referee for this reminder. The S-parameter (scattering parameter) is usually used to characterize the electromagnetic response of a device to RF signal (Pozar D M. Microwave engineering. John Wiley &

Sons (2011)). It takes the input and the output signals at different ports as the matrix elements to measure the transmission and scattering properties of the device. The component $S_{11}(\vec{r}_j, \nu)$ is defined as the reflectivity at site \vec{r}_j for the frequency ν , measured from the input and output signals of the same port 1. Since we only use this component in this study, in order to avoid confusion, we change this notation to $R(\vec{r}_j, \nu)$.

Reviewer A.5: 2. p.3, right column, 2nd paragraph from the top, last sentence: I presume "fidelity of the Landau level" quantifies its flatness? I would suggest to define it explicitly

Reply: We thank the referee for pointing this out. The fidelity is defined as

$$F(m) = \frac{\sum_i D_t(\vec{r}_i, m) \cdot D_e(\vec{r}_i, \nu)}{\sqrt{|\sum_i D_t^2(\vec{r}_i, m)| \cdot |\sum_i D_e^2(\vec{r}_i, \nu)|}}, \quad (\text{R1})$$

where $D_t(\vec{r}_i, m)$ is the theoretical local DOS of position \vec{r}_i and energy m , and $D_e(\vec{r}_i, \nu)$ is the experimental local DOS of position \vec{r}_i and which is integrated at frequency range $\nu \pm 5$ MHz., which is added in the Supplementary Section IV. As pointed by the referee, it can be used as an indicator to characterize the band flatness. However, to experimentally determine the flatness of the band, we measure the ratio of the linewidth of each flatband DOS peak to that of a single resonator. We add this information in the Supplementary Section IV.

Reviewer A.6: 3. Fig. 4: "insets" – I presume the insets are the plots in the middle? It might be better to refer to them as simply "the plots in the middle"?

Reply: We apologize for this misleading description and we follow the advice of the referee to revise the text.

Reply to reviewer B

Reviewer B.1: In the manuscript "Realization of all-band-flat photonic lattices", Yang et al. investigate the experimental realization with microwave resonators of a lattice model with only flat bands, which corresponds to a strained honeycomb lattice beyond the perturbative regime in a Fock-space lattice (FSL). While in FSLs the ad-hoc space-dependent couplings naturally arise via the bosonic statistics of the creation/annihilation operators, here the authors employ the tunability of the platform to exactly match those couplings for the resonators. They observe flat bands and also some properties of the eigenstates (their chirality or phase winding). The manuscript is scientifically sound and the results are convincing.

Reply: We thank the Referee for carefully reading our manuscript and judging that our work is scientifically sound and the results are convincing.

Reviewer B.2: However, I believe that this work is suited to a more specialized journal for the following reasons: 1) on the theoretical side, the authors are realizing a linear system whose physics is relatively simple and known. Besides, many other models with flat bands have been realized over the last years, so I do not find a particular advance here, except that this work provides the realization of yet another model with all bands flat, an extremely interesting example of suppressed transport distinct from AB caging. Thus, I find the results incremental in the respect of many realizations of flat band systems.

Reply: We thank the referee for judging that our result is "an extremely interesting example of suppressed transport distinct from AB caging". However, we respectfully disagree that our advance is incremental and suited to a specialized journal. From the theoretical side, we would like to emphasize that our scheme realizes two-dimensional all-band-flat photonic lattices by mimicking many-body bosonic properties, which is at the interface of photonics, quantum optics, and condensed matter physics, of interest to a broad audience.

All previous methods of realizing flat bands rely on synthesizing effective magnetic field in the lattice. Among these methods, the AB caging usually needs complex spatial and time modulation, while strain engineering are limited in the perturbative regime. Due to the finite size, nonperturbative and edge effect, these methods cannot make all energy bands flat in a finite two-dimensional lattice. Our method, although sharing similarity to the strain engineering (in the middle of the lattice), is essentially based on a photonic emulation of the many-body Jaynes-Cummings model. The coupling strength variation follows the bosonic statistical properties of the creation and annihilation operators, and the range of the modulation goes beyond the perturbative regime of strain engineering, and cannot be attributed to strain-induced effective magnetic effect near the lattice edge. The simplicity of the method can facilitate the application of the scheme in many other photonic systems. This new idea of implementing ABF lattices is free from finite size effect and edge effect and can be generalized to three dimensions, which cannot be easily achieved by using methods based on effective magnetic field. We compared our scheme and previous models that achieve flat bands in Tab. S2 in the Supplementary Material.

Reviewer B.3: 2) on the experimental side, the realization of the model does not seem to require particular challenging steps, techniques, or innovation, but standard engineering methods for this platform. I therefore fail to find a nontrivial experimental advance.

Reply: We apologize for missing the discussion on the experimental advance in our manuscript. First of all, we regard it as an advantage for not involving extremely challenging techniques and engineering methods in our experiment, which is essential for its accessibility. However, to make all bands flat, we do have innovation in our approach for realizing wide tuning range and precise control of the coupling strengths while keeping a narrow linewidth of the bands. We emphasize this point in the revised manuscript.

Reviewer B.4: Besides, while the needed couplings arise naturally with harmonic oscillator systems here they require ad-hoc engineering, thus wondering if there is a need to use this platform if no other advantage (besides the possibility to realize large lattices) is possible. If the authors were able to provide arguments that these results can hardly be seen in most of the other platforms (like photonic waveguides or ring resonators, where large systems and programmable couplings can also be designed with relatively small experimental effort), thus explaining the need or unique advantages of this platform or if they demonstrated interesting nonlinear effects on such flat bands, this would be a quite different story.

Reply: We thank the Referee for asking us to compare our experiment with the existing platforms, which greatly helped us to improve the presentation. First, we need to emphasize that the realization of the original Fock-state lattices, where the flatbands arise from the naturally determined coupling strengths between the Fock states, needs strong coupling between an atom and several cavities. Up to now, it has only been realized in superconducting circuits

at extremely low temperature (10 mK). Our current research borrows the physics from Fock-state lattices to solve the problem of realizing all-band-flat spectra in photonic lattices, which are of fundamental interest for caging and nonlinear effects and have promising applications in dispersionless imaging. The significance of all-band-flat photonic lattices has been evident from the effort in realizing these lattices in various platforms. Our scheme is the first to realize all-band-flat spectra in finite lattices, which has never been implemented in those platforms. The strain engineering is similar to our approach but in the perturbative regime such that not all bands are flat (see Ref. [51]). With this being said, we can only list the difficulties in realizing such a scheme in photonic waveguides or ring resonators, without claiming the impossibility of such a realization. At the same time, it is evident that these difficulties have been circumvented in our platform.

For waveguides and ring resonators, the coupling strength modulation has been achieved by changing the distances between them. However, to individually and precisely control the coupling strengths in a lattice of waveguides by changing their relative distances requires sophisticated deformation of the geometry of the lattice, which is a formidable (if not impossible) task when the tuning range is nonperturbative and the lattice is large. In changing the distances between waveguides, it is also hard to avoid unwanted coupling between next-nearest-neighbor waveguides. In our lattice, the coupling strengths are tuned without changing the distance between the resonators, but by changing the widths of the channels connecting them, such that precise ad-hoc control of the coupling strengths is achieved without deforming the lattice geometry or inducing unwanted couplings.

Such a method also allows us to generalize the lattice to three dimensions with all-band-flat spectra, which cannot be achieved with waveguides or dielectric resonators. We demonstrate such a 3D all-band-flat unit and its simulated energy spectra in the Supplementary Section VI. By using diodes we can introduce and engineer the nonlinearity in the resonators. We demonstrate the effect of such nonlinearity in the Supplementary Section VII. The channel can also be filled with medium or active devices, which can be modulated with time, generating effective magnetic field that breaks the time-reversal symmetry. We demonstrate the scheme of introducing dynamic coupling modulation in the Supplementary Section VIII.

Reviewer B.5: Here a couple of comments: - I could not find a fully clear explanation of how the coupling strength is experimentally tuned as \sqrt{i} . Is it done by changing the widths d_i or by other means? I suppose that the sentence "Three openings with individually designed widths.." below eq. 2 is the answer to my doubt. If that's the case, I suggest to be explicit and indicate that the individual design of the openings is designed to provide the \sqrt{i} couplings in order to avoid that a reader who is not completely familiar with the platform has to deduce this on their own.

Reply: We thank the Referee for this kind suggestion and we apologize for missing to explain this extremely important point in our manuscript. Indeed, the coupling strengths are tuned by changing the widths d_i , and each resonator is coupled to 3 adjacent resonators. We added the description on this point.

Reviewer B.6: - At the end of the manuscript, the authors indicate the possibility of exploring nonlinear effects by adding a nonlinear medium. As the authors are not citing any reference for this, is it experimentally possible and can this lead to sizeable nonlinear effects? Can they cite experimental literature showing such potential of nonlinear physics with this platform.

Reply: We thank the Referee for the suggestion on elaborating the experimental possibility of nonlinear effect in our model. In the revised manuscript, we cite previous works on adding nonlinearity to the microwave resonator. We can use varactor diodes to realize tunable nonlinearity, which allows for Kerr nonlinearity, with self-induced frequency shifts up to 100 MHz. In the experiment, we can weld the varactors to the rod in the middle of the resonator. The following table (also in Supplementary Table. S3) supports the feasibility of our proposal.

Table I. Nonlinearity in microwave resonators.

Article	System	Nonlinear element	Nonlinear type	Self-reduced frequency shift / coupling strength (ω_0)	Working frequency
[74]Phys. Rev. Lett. 121.163901 (2018)	resonators	varactor diodes	Kerr effect	0-0.06 / 0.03	1500 MHz
[75]Nat. Electron. 1, 178 (2018).	electronic circuit	nonlinear varactor diodes	Kerr effect	0.135-0.02 / 0.055	100 MHz
[80]Appl. Phys. Lett. 109, 111904 (2016)	meta-material	doubly resonant coupled split-ring resonatorâs (SRR)	Second-harmonic generation (SHG)	-	2000 MHz
[81]Nat. Commun. 10, 1102 (2019)	electronic circuit	nonlinear capacitor	Second-harmonic generation (SHG)	-	19 MHz

Reviewer B.7: - The authors mention nonlinear caging as an outlook. However, caging (even in the nonlinear regime) typically involves models that have all-bands-flat dispersions due to interference in the linear regime and special connectivities of the lattice, namely different sublattices having different connectivities. I, therefore, do not see why the model studied in this work would lead to non-linear caging effects, in the same way as Landau levels are not known, to the best of my knowledge, to have the property of non-linear caging. If the authors are making a conjecture, or have in mind literature in this direction, they should explain what reasonings have they made to believe that such phenomenology of non-linear caging would occur here. Otherwise, they should remove a claim of nonlinear caging. Besides, nonlinear caging was initially studied in the following articles that the authors are missing from their references:

Nonlinear symmetry breaking of Aharonov-Bohm cages. Goran Gligorić, Petra P. Beličev, Daniel Leykam, and Aleksandra Maluckov Phys. Rev. A 99, 013826 - Published 15 January 2019

Nonlinear dynamics of Aharonov-Bohm cages. Marco Di Liberto, Seabrata Mukherjee, and Nathan Goldman Phys. Rev. A 100, 043829 - Published 22 October 2019

Reply: We thank the Referee for this comment, which gives us a chance to elaborate on this outlook. The nonlinear caging can be observed in ABF lattices that at the same time possess nonlinearities (or many-body interactions). Indeed these effects have been investigated in complex lattices that have different connectivities for each sublattice. It is also true that Landau levels are not proven to host nonlinear caging effect. However, we would like to emphasize that our lattices are by no means Landau-level systems. Only in the middle of the lattice (in real space) and near the Dirac points (in momentum space) can the energy spectra be approximately attributed to strain induced Landau levels. Therefore, by introducing nonlinearity in the current lattices, it is possible to observe nonlinear localization and other interesting effect, which is under our current investigation. We added some simulation results on the nonlinear breathing dynamics in the Supplementary Section IX.

Reviewer B.8: Is there any other interesting nonlinear effect that could be explored in this model if caging is ruled out? I think the outlook part is a bit vague and may benefit from a clearer perspective on the impact of this work.

Reply: Besides the nonlinear effect due to interactions between photons, we can also add other nonlinearities that are only possible for bosons. One possibility is the second harmonic nonlinearity, $\chi^{(2)}$. The flat bands, which can have high density of states in a narrow spectra region, can enhance the second harmonic generation (Opt. Lett. 47, 2326 (2022); Nanophotonics 12, 4009-4016 (2023)). Such enhancement by flat bands in nonlinear photonic lattices is still a new direction yet to be investigated. Other interesting effects in all-band-flat photon lattices include photonic soliton states, which crucially depend on the flatness of the bands (Nat. Photon. 14, 663 (2020)). We added a brief discussion on this perspective in the conclusion.

Reviewer B.9: - I take the opportunity to stress that tight-binding models with couplings dependent on the \sqrt{n} , where n is the fock-state occupation, may arise in general setups that exploit harmonic oscillator states as a synthetic dimension, besides the JC models that the authors are referring to. For example, here are some relevant works in this direction that the authors could cite

Synthetic dimensions for cold atoms from shaking a harmonic trap. Hannah M. Price, Tomoki Ozawa, and Nathan Goldman Phys. Rev. A 95, 023607 Published 9 February 2017

Quantized Hall Conductance of a Single Atomic Wire: A Proposal Based on Synthetic Dimensions. G. Salerno, H. M. Price, M. Lebrat, S. Häusler, T. Esslinger, L. Corman, J.-P. Brantut, and N. Goldman Phys. Rev. X 9, 041001 Published 1 October 2019

Photonic topological insulator in synthetic dimensions. Eran Lustig, Steffen Weimann, Yonatan Plotnik, Yaakov Lumer, Miguel A. Bandres, Alexander Szameit & Mordechai Segev Nature volume 567, pages356-360 (2019)

Reply: We thank the Referee for the suggestion. We have cited related references in the proper places in the main text.

REVIEWERS' COMMENTS

Reviewer #1 (Remarks to the Author):

I believe the authors properly addressed all of my questions/remarks as well as those of the other referee and the manuscript is now suitable for publication.

I would like to reiterate the reasons for publication:

1. While there is now indeed a significant set of experimental realisations of flatbands in various settings, realisations of all bands flat are rare -- I am only aware of two:

New J. Phys. 22 (2020) 013023 and arxiv:2303.02170, and both are relying on Aharonov-Bohm caging.

A different realisation of all bands flat is therefore welcome, and can serve as a platform for studying various perturbations, e.g. non-linearities or disorder.

2. The realisation in the present work is straightforward and relatively not challenging, as pointed out by the authors and the other referee.

I think this is a benefit in this specific context -- achieving all bands flat.

I would also like to add a comment to the remark of the other referee:

"However, caging (even in the nonlinear regime) typically involves models that have all-bands-flat dispersions due to interference in the linear regime and special connectivities of the lattice, namely different sublattices having different connectivities."

I believe that nonlinear caging does not require different connectivities for different sublattices.

One example is the nonlinear Creutz ladder, see Phys. Rev. B 104, 085131 (2021).

Reviewer #2 (Remarks to the Author):

Dear editors,

I have reviewed the new version of the manuscript "Realization of all-band-flat photonic lattices", and I found the rebuttal only in part convincing. More specifically, I understood that the platform used here is

quite effective to realize Fock-state lattices, which may be difficult in other systems based on tuning hopping couplings by distance. What I cannot still clarify is what this experimental realization will truly bring as innovative and ground breaking as compared to other realizations of all-flat band lattices.

Besides, I disagree with the statement "All previous methods of realizing flat bands rely on synthesizing effective magnetic field in the lattice. Among these methods, the AB caging usually needs complex spatial and time modulation". My disagreement on the previous statement is that AB caging typically requires what is known as π -flux, which can be obtained without complex space or time modulation but, for example, by changing the hopping sign in certain specific lattice bonds. This is something relatively simple to realize, for example in the diamond chain, and it was shown in 1D by Szameit's group (A square-root topological insulator with non-quantized indices realized with photonic Aharonov-Bohm cages, Mark Kremer, Ioannis Petrides, Eric Meyer, Matthias Heinrich, Oded Zilberberg & Alexander Szameit, Nature Communications volume 11, Article number: 907 (2020)) by adding one extra waveguide between certain bonds. In 2D, one can obtain an all-flat-band model of the AB caging type by using the dice lattice geometry with π -flux (see Moller and Cooper PRL 108, 045306 (2012)), assuming a practical realization can be pursued respecting all crystal symmetries (which I do not know if truly possible).

The main advance that I see in the manuscript is the realization in a controllable manner of a large set of flat band states at different energies but I am not convinced this is enough for a publication in Nature Communications. Besides the fact that the work connects with notions of other fields (like quantum optics and JC model), what I fail to see is the real groundbreaking novelty that these results are bringing. The linear effects are not so different than AB caging (which in principle have been already observed in multiple setups without a complicated experimental effort, ranging from waveguides using s-modes/p-modes/OAM modes to cold atoms with synthetic dimensions). A non-exhaustive list of AB caging experiments is provided here:

- Experimental Observation of Aharonov-Bohm Cages in Photonic Lattices, Phys. Rev. Lett. 121, 075502 – Published 16 August 2018
- A square-root topological insulator with non-quantized indices realized with photonic Aharonov-Bohm cages, Nature Communications volume 11, Article number: 907 (2020)
- Artificial gauge field switching using orbital angular momentum modes in optical waveguides, Light: Science & Applications volume 9, Article number: 150 (2020)
- Aharonov-Bohm Caging and Inverse Anderson Transition in Ultracold Atoms, Phys. Rev. Lett. 129, 220403 – Published 22 November 2022
- Controlled Transport Based on Multiorbital Aharonov-Bohm Photonic Caging, Phys. Rev. Lett. 128, 256602 – Published 23 June 2022

The difference with AB caging is, from I can see in the manuscript, that the construction of the manuscript brings the possibility to have many modes at many different frequencies that are still localized whereas AB caging typically involve all-flat-bands with just few frequencies, but is not bringing qualitatively new phenomena, in my opinion. In this respect, I fail to see the groundbreaking impact of this difference.

Showing an experimental part with quite unexplored phenomenology, for example originating from unconventional/unobserved nonlinear effects or some other peculiar effects of this system besides localization, would have been a quite different story. However, again I fail to see what is different or richer with respect to many other works that have experimentally investigated nonlinear dynamics originating from flat bands. If there were unique and distinct nonlinear phenomena (or other effects) occurring in fock-space lattices that have not been seen or possible in other flat-band contexts (nonlinear mode breathing is a quite established effect), and are enabled by this platform then it would be a quite different perspective.

With all this said, I find the results sound and interesting but publishable in a journal different from Nat. Communications.

Reply to reviewer A

Reviewer A.1: I believe the authors properly addressed all of my questions/remarks as well as those of the other referee and the manuscript is now suitable for publication.

Reply: We thank the Referee for saying that we have properly addressed both Referees' questions and supporting the publication of our manuscript.

Reviewer A.2: I would like to reiterate the reasons for publication:

1. While there is now indeed a significant set of experimental realisations of flatbands in various settings, realisations of all bands flat are rare – I am only aware of two:

New J. Phys. 22 (2020) 013023 and arxiv:2303.02170, and both are relying on Aharonov-Bohm caging. A different realisation of all bands flat is therefore welcome, and can serve as a platform for studying various perturbations, e.g. non-linearities or disorders.

2. The realisation in the present work is straightforward and relatively not challenging, as pointed out by the authors and the other referee. I think this is a benefit in this specific context – achieving all bands flat.

Reply: We thank the Referee's comment on the importance of our works.

Reviewer A.3: I would also like to add a comment to the remark of the other referee: "However, caging (even in the nonlinear regime) typically involves models that have all-bands-flat dispersions due to interference in the linear regime and special connectivities of the lattice, namely different sublattices having different connectivities." I believe that nonlinear caging does not require different connectivities for different sublattices. One example is the nonlinear Creutz ladder, see Phys. Rev. B 104, 085131 (2021).

Reply: We thank the Referee for this comment.

Reply to reviewer B

Reviewer B.1: I have reviewed the new version of the manuscript "Realization of all-band-flat photonic lattices", and I found the rebuttal only in part convincing. More specifically, I understood that the platform used here is quite effective to realize Fock-state lattices, which may be difficult in other systems based on tuning hopping couplings by distance. What I cannot still clarify is what this experimental realization will truly bring as innovative and ground breaking as compared to other realizations of all-flat band lattices.

Reply: In the last response, we have compared our approach with other approaches and demonstrated the advantage of the current method. Our experiment is one of the very few experiments that actually realized all-band-flat lattices (with unprecedented and scalable lattice size), as pointed out by Reviewer A, who also added that "A different realisation of all bands flat is therefore welcome, and can serve as a platform for studying various perturbations, e.g. non-linearities or disorders."

Reviewer B.2: Besides, I disagree with the statement "All previous methods of realizing flat bands rely on synthesizing effective magnetic field in the lattice. Among these methods, the AB caging usually needs complex spatial and time modulation". My disagreement on the previous statement is that AB caging typically requires what is known as pi-flux, which can be obtained without complex space or time modulation but, for example, by changing the hopping sign in certain specific lattice bonds. This is something relatively simple to realize, for example in the diamond chain, and it was shown in 1D by Szameit's group (A square-root topological insulator with non-quantized indices realized with photonic Aharonov-Bohm cages, Mark Kremer, Ioannis Petrides, Eric Meyer, Matthias Heinrich, Oded Zilberberg & Alexander Szameit, Nature Communications volume 11, Article number: 907 (2020)) by adding one extra waveguide between certain bonds. In 2D, one can obtain an all-flat-band model of the AB caging type by using the dice lattice geometry with pi-flux (see Moller and Cooper PRL 108, 045306 (2012)), assuming a practical realization can be pursued respecting all crystal symmetries (which I do not know if truly possible).

Reply: The two papers listed by the Referee only show how difficult to realize flat bands using AB caging effect. While the experimental work in [Nat. Commun. 11, 907 (2020)] is in a 1D lattice, the 2D lattice paper [PRL 108, 045306 (2012)] is only a theoretical proposal. It was more than 10 years ago and no experimental realization has been demonstrated, which itself demonstrates the difficulty. The Referee B also seemed unsure of the feasibility.

Reviewer B.3: The main advance that I see in the manuscript is the realization in a controllable manner of a large set of flat band states at different energies but I am not convinced this is enough for a publication in Nature Communications. Besides the fact that the work connects with notions of other fields (like quantum optics and JC model), what I fail to see is the real groundbreaking novelty that these results are bringing. The linear effects are not so different than AB caging (which in principle have been already observed in multiple setups without a complicated experimental effort, ranging from waveguides using s-modes/p-modes/OAM modes to cold atoms with synthetic dimensions). A non-exhaustive list of AB caging experiments is provided here:

- Experimental Observation of Aharonov-Bohm Cages in Photonic Lattices, Phys. Rev. Lett. 121, 075502 - Published 16 August 2018
- A square-root topological insulator with non-quantized indices realized with photonic Aharonov-Bohm cages, Nature Communications volume 11, Article number: 907 (2020)
- Artificial gauge field switching using orbital angular momentum modes in optical waveguides, Light: Science & Applications volume 9, Article number: 150 (2020)
- Aharonov-Bohm Caging and Inverse Anderson Transition in Ultracold Atoms, Phys. Rev. Lett. 129, 220403 - Published 22 November 2022
- Controlled Transport Based on Multiorbital Aharonov-Bohm Photonic Caging, Phys. Rev. Lett. 128, 256602 - Published 23 June 2022

Reply: Again, all the listed references demonstrate that our advance is significant. These references only involve with the same 1D diamond lattice structure with limited lattice size. These lattices have edge states which do not belong to the flat bands. Precisely speaking they have not realized all-band-flat lattices due to the edge effect. Our 2D realization of all-band-flat lattices with scalable lattice size significantly enriches the physics that can be explored in flat-band lattices, as shown in our supplementary materials.

Reviewer B.4: The difference with AB caging is, from I can see in the manuscript, that the construction of the manuscript brings the possibility to have many modes at many different frequencies that are still localized whereas AB caging typically involve all-flat-bands with just few frequencies, but is not bringing qualitatively new phenomena, in my opinion. In this respect, I fail to see the groundbreaking impact of this difference.

Reply: The Referee is right that our lattices have more flat bands than in the AB cages. As we have shown in our paper, different flatbands have different spatial distributions in the lattices, which gives us the freedom in exciting the

target flatband states by controlling the excitation location, frequency and chirality. Therefore, we can use the full energy bands for dispersionless image transmission and other applications. The nonlinear dynamics are also richer than that of the AB caging lattices, as shown in the supplementary materials. Such controllability is absent in AB caging lattices.

Reviewer B.5: Showing an experimental part with quite unexplored phenomenology, for example originating from unconventional/unobserved nonlinear effects or some other peculiar effects of this system besides localization, would have been a quite different story. However, again I fail to see what is different or richer with respect to many other works that have experimentally investigated nonlinear dynamics originating from flat bands. If there were unique and distinct nonlinear phenomena (or other effects) occurring in fock-space lattices that have not been seen or possible in other flat-band contexts (nonlinear mode breathing is a quite established effect), and are enabled by this platform then it would be a quite different perspective.

With all this said, I find the results sound and interesting but publishable in a journal different from Nat. Communications.

Reply: New nonlinear effect is a very promising direction that we are currently pursuing. However, it is beyond the scope and theme of the current paper, i.e., a concrete demonstration of all-band-flat energy spectra in 2D lattices, with a scalable lattice size.